# Genome of the endangered eastern quoll (*Dasyurus viverrinus*) reveals signatures of historical decline and pelage color evolution

Gabrielle A. Hartley[1], Stephen R. Frankenberg [2], Natasha M. Robinson [3], Anna J. MacDonald [4,12], Rodrigo K. Hamede[5], Christopher P. Burridge [5], Menna E. Jones[5], Tim Faulkner[6], Hayley Shute[6], Karrie Rose [7], Rob Brewster[8], Rachel J. O'Neill [1,9], Marilyn B. Renfree [2], Andrew J. Pask [2,10] & Charles Y. Feigin [2,11] ✉

The eastern quoll (*Dasyurus viverrinus*) is an endangered marsupial native to Australia. Since the extirpation of its mainland populations in the 20th century, wild eastern quolls have been restricted to two islands at the southern end of their historical range. Eastern quolls are the subject of captive breeding programs and attempts have been made to re-establish a population in mainland Australia. However, few resources currently exist to guide the genetic management of this species. Here, we generated a reference genome for the eastern quoll with gene annotations supported by multi-tissue transcriptomes. Our assembly is among the most complete marsupial genomes currently available. Using this assembly, we infer the species' demographic history, identifying potential evidence of a long-term decline beginning in the late Pleistocene. Finally, we identify a deletion at the *ASIP* locus that likely underpins pelage color differences between the eastern quoll and the closely related Tasmanian devil (*Sarcophilus harrisii*).

The eastern quoll (*Dasyurus viverrinus*; Fig. 1a) is a marsupial carnivore native to Australia[1]. A nocturnal mesopredator, the eastern quoll's diet primarily consists of small mammals, reptile and invertebrate prey, as well as opportunistic scavenging on the remains of larger species[2]. The eastern quoll is morphologically, behaviorally, and ecologically distinct from the larger, parapatric spotted-tailed quoll (*Dasyurus maculatus*)[3]. Eastern quolls give birth to exceptionally altricial young that are comparable to mid-gestation fetuses of eutherian mammals[4,5]. Indeed, neonates have structurally immature lungs and perform as much as 95% of their gas exchange through their skin[6–8]. Like other marsupials, their young travel from the birth canal into their mother's pouch on the day of birth and complete much of their development *ex utero*[1,9].

Despite having once ranged across much of southeastern Australia, eastern quolls were likely extirpated from the mainland by the late 20th century[10]. Today, natural populations of this species are restricted to the state of Tasmania on two islands (the Tasmanian main island and Bruny Island; Fig. 1b), which lie at the southern end of their historical range. Moreover, several Tasmanian eastern quoll populations have also undergone significant and ongoing population declines in recent decades and the species was declared 'Endangered' by IUCN in 2016[11–13]. Suggested causes of decline include invasive predators and climatic fluctuation[14,15]. The eastern quoll has been flagged as one of 20 priority mammal species in the Australian Government's Threatened Species Strategy, with conservation efforts including population supplementation studies, an extensive captive

[1]Institute for Systems Genomics, University of Connecticut, Storrs, CT 06269, USA. [2]School of BioSciences, The University of Melbourne, Melbourne, VIC 3010, Australia. [3]Fenner School of Environment & Society, Australian National University, Canberra, ACT 2601, Australia. [4]Research School of Biology, Australian National University, Canberra, ACT 2601, Australia. [5]School of Natural Sciences, University of Tasmania, Hobart, TAS 7005, Australia. [6]Australian Reptile Park & Aussie Ark, Somersby, NSW 2250, Australia. [7]Australian Registry of Wildlife Health, Taronga Conservation Society Australia, Mosman, NSW 2088, Australia. [8]WWF-Australia, PO Box 528 Sydney, NSW 2001, Australia. [9]Department of Molecular and Cell Biology, University of Connecticut, Storrs, CT 06269, USA. [10]Department of Sciences, Museums Victoria, Carlton, VIC 3053, Australia. [11]Department of Environment and Genetics, La Trobe University, Bundoora, VIC 3086, Australia. [12]Present address: Australian Antarctic Division, Department of Climate Change, Energy, the Environment and Water, Kingston, TAS 7050, Australia. ✉e-mail: c.feigin@latrobe.edu.au

**Fig. 1 | Appearance and distribution of the eastern quoll. a** Photograph of an adult eastern quoll (photo credit Brett Vercoe). **b** Map of the state of Tasmania, showing 50 years of eastern quoll sightings across the Tasmanian main island and Bruny Island recorded in the Tasmanian Natural Values Atlas. Individual sightings are shown as blue dots.

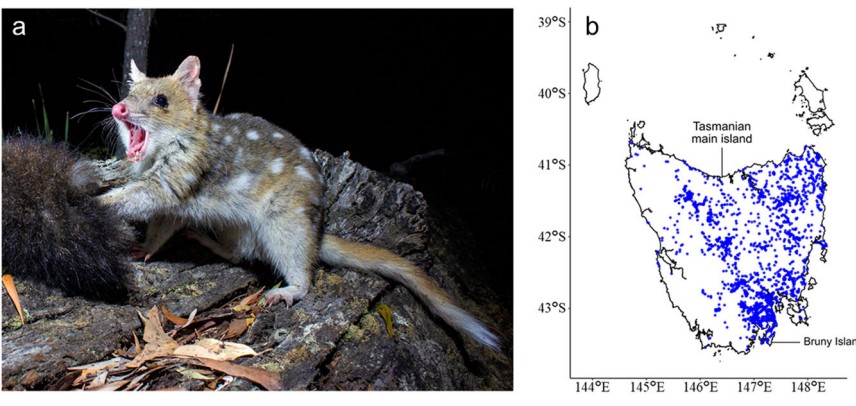

breeding program spanning multiple sanctuaries, and fenced safe havens free of cats and foxes[16,17]. The species has also recently been the subject of an ambitious project aimed at re-establishing wild populations in parts of its former mainland range, with two pilot releases of captive-bred animals in Booderee National Park on the south coast of New South Wales[18,19].

While previous population-level research has gleaned some insights into population structure and provided evidence of limited genetic diversity among Tasmanian eastern quolls using low-resolution microsatellite markers, the overall paucity of genome-scale data and resources presents a barrier to effective conservation genetic management of the species[20]. Furthermore, as a member of the family Dasyuridae the eastern quoll represents a valuable point of comparison for evolutionary genomic studies with related marsupials, such as the Tasmanian devil. Here, we present a high-quality eastern quoll reference genome and conduct genomic studies of its demographic history and the molecular basis of pigment variation in dasyurids.

## Results and discussion
### The eastern quoll reference genome
Eastern quoll samples were collected opportunistically during the post-mortem examination of an adult female, including several tissues which were frozen and live toe pad which was used to isolate primary fibroblasts in culture. Using these, we generated a chromosome-scale, de novo reference genome by assembling ~97.68 gigabases (Gb) of Pacific Biosciences high-fidelity (HiFi) long-reads and ~126 Gb of Omni-C long-range chromatin contact data. We called the resulting assembly DasViv_v1.0.

The assembly size was ~3.14 Gb, comparable to that of related marsupials, and scaffolds were nearly free of internal gaps (Table 1)[21,22]. The assembly was composed of only 76 scaffolds, of which the seven largest corresponded to the conserved karyotype found across all known dasyurids: six autosomes plus the X chromosome (Fig. 2a)[23–26]. Together, the seven chromosome-scale scaffolds accounted for 99.34% of the total assembly size. Homology between eastern quoll chromosome-scale scaffolds and those of

the related Tasmanian devil (*Sarcophilus harrisii*) and yellow-footed antechinus (*Antechinus flavipes*) was confirmed by the high overlap of orthologous gene annotations ( > 95%) and their similar total lengths (Supplementary Table 1)[21,22]. Recovery of complete single-copy mammalian BUSCOs (Benchmarking Universal Single-Copy Orthologs) was 92.2%, second only to the koala (*Phascolarctos cinereus*) among marsupial reference genomes currently available on NCBI (Fig. 2b, Supplementary Table 2). Moreover, rates of duplicated and fragmented BUSCO genes were low (1.3% and 1.2%, respectively), reinforcing the completeness and integrity of our assembly. Given the high quality of the assembly, this resource represents a valuable tool for conservation which can facilitate variant identification and the design of efficient population-level sampling methods (e.g. SNP panel design), as well as evolutionary genomic studies.

### Genome annotation
To accompany our assembly, we generated gene annotations by combining evidence from transcriptome data generated from five tissues (Supplementary Table 3), homologous proteins from related marsupials (Supplementary Table 4), as well as ab initio predictions. In total, we generated 29,622 gene models (Fig. 3a)[21,22,27–29].

We also annotated and characterized the repeat content of the eastern quoll genome using RepeatMasker[30]. In total, ~1.476 gigabases were masked as repetitive (47.2% of the assembly), comprising mainly LINEs (30.61%) and SINEs (8.93%; Fig. 3b). L1 repeats constituted the most abundant LINEs in the assembly (20.15%) and MIRs constituted the most abundant SINEs (7.67%). Notably, we identified a small fraction of bases (7906 in total) corresponding to putative KERV elements, an endogenous retrovirus that has undergone radical expansion in the kangaroo genus *Macropus* (Supplementary Table 5)[31].

### Demographic history
Australia has experienced substantial ecological changes over recent geological epochs. For instance, the late Pleistocene saw extensive climatic changes, the periodic isolation of Tasmania from the mainland, and the arrival and dispersal of humans into Australia[32–34]. This period also marked the extinction of the remaining megafauna (terrestrial vertebrates >40 kilograms)[35]. However, the effects that these potential historical stressors may have had on eastern quoll populations are poorly explored. Therefore, we next sought to examine historical trends in eastern quoll effective population size ($N_e$) with multiple sequentially Markovian coalescent (MSMC) analysis, using a mutation rate for the closely-related Tasmanian devil measured from parent-offspring trios and a generation time of 2 years[36,37].

Our analysis indicated long-term decline in $N_e$, comprised of two phases (Fig. 4a). The first of the inferred decline phases is indicated to have begun ~300 thousand years ago (kya) and continued until approximately 70 kya. However, recent work on the properties of MSMC analysis in the related Tasmanian devil suggest that accuracy

## Table 1 | DasViv_v10 Assembly Metrics

| Metric | Contigs | Scaffolds |
|---|---|---|
| Total Length | 3139.18 Mb | 3139.22 Mb |
| Max Length | 48.42 Mb | 714.54 Mb |
| Number | 507 | 76 |
| N50 | 13.75 Mb | 628.49 Mb |
| N90 | 8.72 Mb | 468.98 Mb |
| G + C % | - | 36.19% |
| Gap % | - | 0.001% |

Basic metrics related to contig, scaffold and overall assembly of the eastern quoll reference genome.

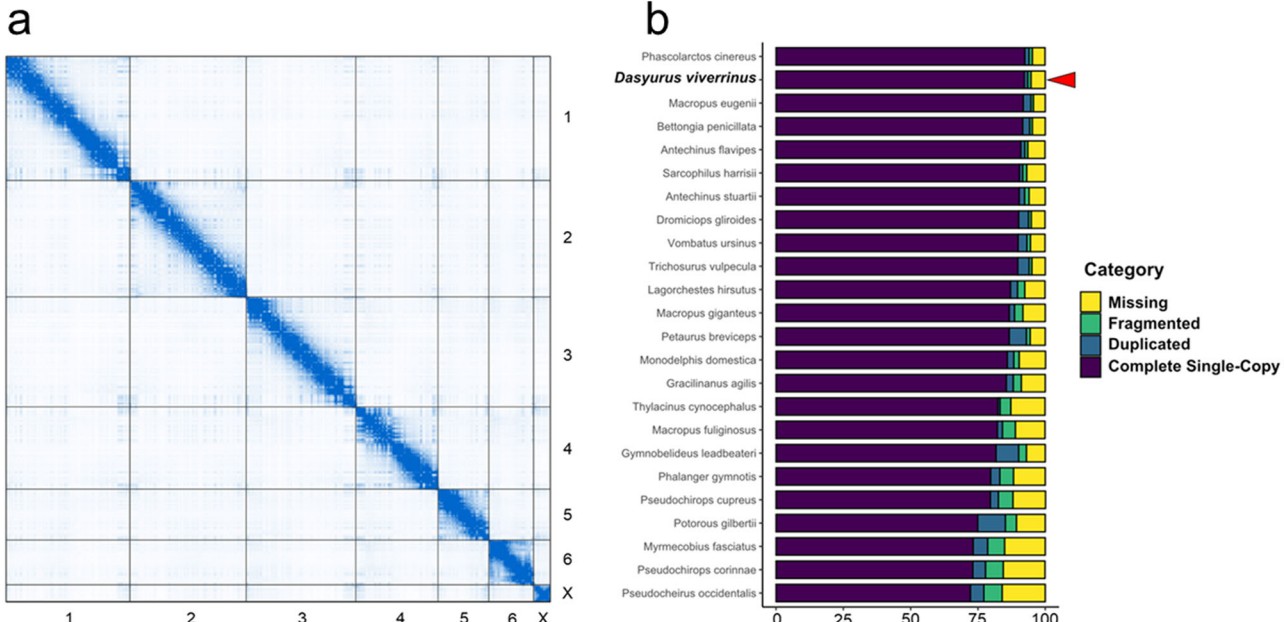

**Fig. 2 | Assessment of genome integrity and completeness. a** Contact map of Omni-C data against chromosome-scale scaffolds in the eastern quoll assembly. The enrichment level of chromosomal contacts is shown as blue pixel intensity **b** Stacked bar plot comparing the recovery of mammalian BUSCO genes. Recovery in the eastern quoll is among the highest for sequenced marsupials, with low rates of duplicated or fragmented orthologs.

**Fig. 3 | Gene and repeat annotation. a** Heat map illustrating the density of annotated genes across eastern quoll chromosome-scale scaffolds. **b** Bar plot showing the distribution of annotated repeats by class.

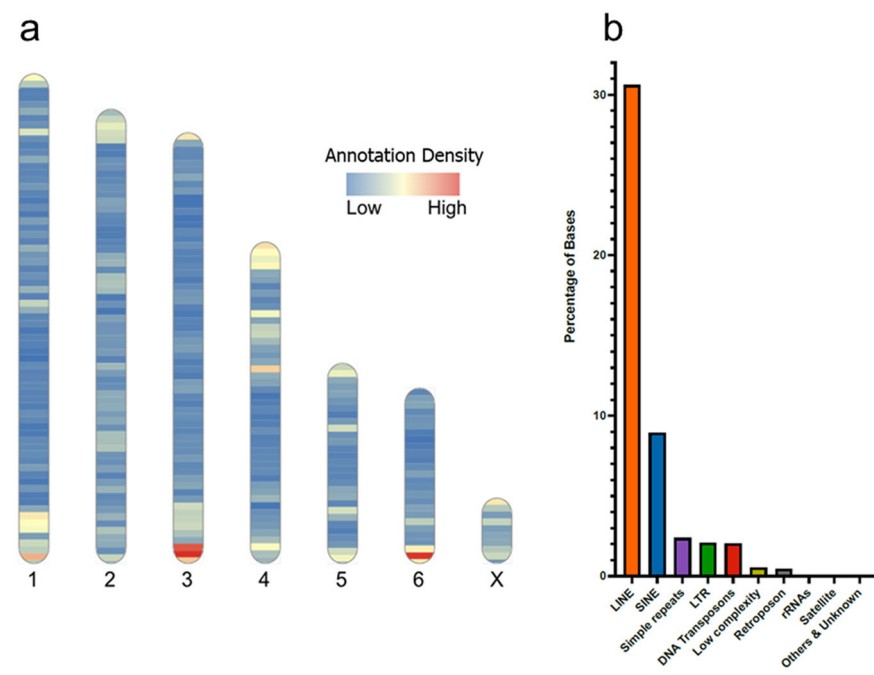

can decline markedly during very recent and very ancient periods (less than 1000 and greater than 100,000 generations ago, respectively)[38]. Thus, we are cautious in over-interpreting the precise start of this decline phase, which falls outside of this range in our analysis (~2–200 kya). The more recent inferred decline began approximately 28 kya and extended into the Holocene (Fig. 4a). The arrival of humans in Tasmania ~40 kya seems an unlikely candidate to have driven this decline, as they co-existed with eastern quolls for some 12 kyr with apparently little impact on $N_e$[34]. Curiously, our analyses indicate that both declines may have initiated during periods in which Tasmania was connected to mainland Australia due to lowered sea levels[39], despite the potential for increased gene flow between Tasmanian and mainland

populations. It is possible that unfavorable climates during these colder periods created suboptimal environmental conditions in Tasmania and it is unclear whether the Bassian Plain (the land bridge connecting Tasmania and mainland Australia) environments were conducive to gene flow. Indeed, bare-nosed wombat (*Vombatus ursinus*) populations with past territorial connections via the Bassian Plain (i.e. on Tasmania, Flinders Island, and mainland Australia) appear to have maintained genetic isolation[40]. Future resequencing studies of eastern quolls from Tasmanian, Bruny Island, and preserved historical mainland specimens may provide further insights into historical gene flow and population structure, which can confound MSMC and related analyses[41].

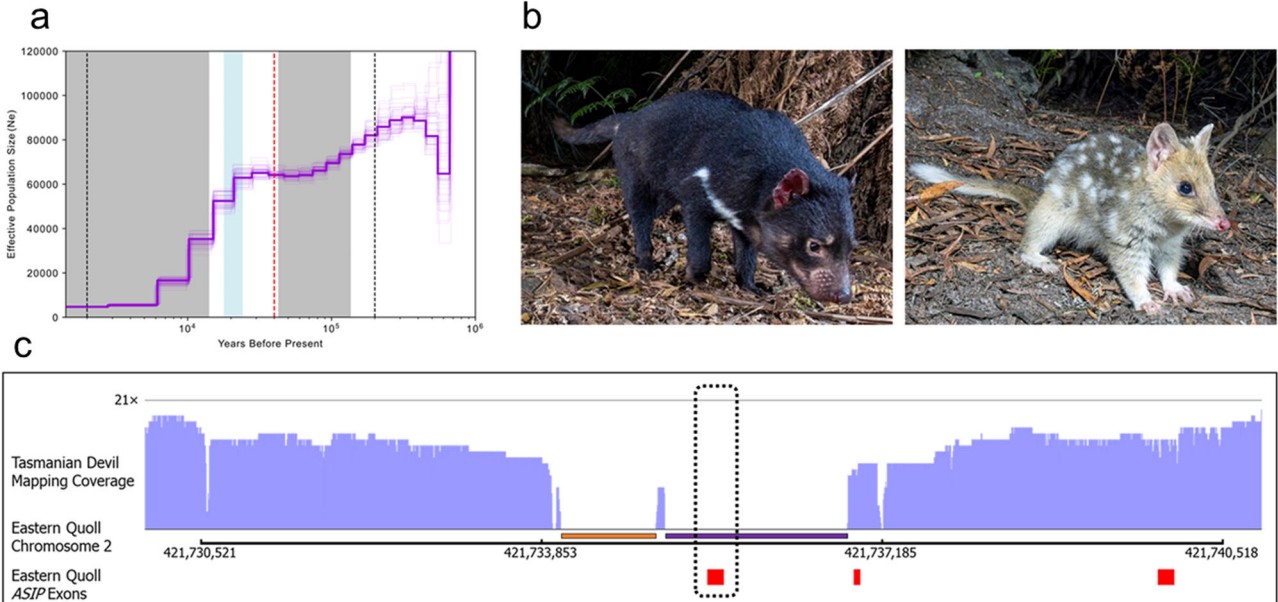

**Fig. 4 | Demography and comparative genomics of pigmentation. a** Step plot illustrating inferred changes in eastern quoll effective population size ($N_e$) over time. Grey regions indicate periods in which Tasmania was separated from mainland Australia (~135–43 kya and ~14 kya to present). The blue region indicates the coldest period of the Last Glacial Maximum (~24–18 kya). The dashed red line represents the arrival of humans in Tasmania (~40 kya). Area between the black dashed lines at 2000 and 200,000 years represent the window within which MSMC analysis is expected to be accurate based on previous studies (1–100 thousand generations before present)[38]. **b** Photographs comparing pelage color of the Tasmanian devil and eastern quoll (photo credits Brett Vercoe). **c** Mapping coverage of Tasmanian devil long reads (in light purple) across the *ASIP* locus in the eastern quoll genome. Exons of the eastern quoll *ASIP* gene are shown as red blocks. The deletion region is indicated by the absence of mapped Tasmanian devil reads underlined by a purple bar and encompassing exon 1 (dashed box) of *ASIP*. The putative eastern quoll insertion is underlined by an orange bar, upstream of the Tasmanian devil deletion.

## Comparative genomics of pigmentation

The closest living relative to the quolls (genus *Dasyurus*) is the Tasmanian devil. Notably, quolls and Tasmanian devils differ markedly in their coat color and patterning. The quoll background coat color consists of brown dorsal fur, punctuated by spots of white fur and white-to-yellow ventral fur. In contrast, Tasmanian devils have nearly uniform brown/black fur on both dorsum and ventrum, with many individuals bearing unpigmented white patches on the chest, shoulders and/or base of the tail (Fig. 4b). The color of the background dorsal fur of eastern quolls is produced by alternating bands of yellow pheomelanin and dark brown eumelanin in individual hair shafts, a pattern called "agouti"[42]. The agouti pattern, which is common among diverse mammals and likely ancestral among dasyurid marsupials, is known to be regulated in part by the interactions of two key proteins, the melanocortin 1 receptor (MC1R) which promotes eumelanin production and agouti signaling peptide (ASIP), which antagonizes MC1R leading to pheomelanin synthesis[43]. Melanistic morphs of many animals have been shown to be caused by loss-of-function (LOF) mutations in the *ASIP* gene, including multiple independent polymorphisms among *Peromyscus* mice and in wild cats[44,45]. Therefore, we hypothesized that the dark, eumelanin-bearing hair in Tasmanian devils might therefore have evolved through a comparable mechanism.

We first sought to compare ASIP orthologs from our eastern quoll genome (DasViv_v1.0) and the reference genomes of the Tasmanian devil (mSarHar1.11) with those from several other dasyuromorph species by aligning their coding sequences (Supplementary Data 1 and Supplementary Table 6)[21,22,46–48]. Notably, while extracting orthologs from each reference genome, we failed to identify the first exon of ASIP from mSarHar1.11, despite the assembly being of high-quality and containing few gaps[22]. To rule out the possibility that this region is present in the Tasmanian devil genome, but not reliably assembled during the construction of mSarHar.1.11, we aligned the nanopore long reads used to produce this genome against our eastern quoll genome. A histogram of read coverage showed two regions in which no Tasmanian devil reads mapped (Fig. 4c). Among these, the second

region (~1.8 kb long) encompassed the entirety of *ASIP* exon 1. Together, these observations indicated the presence of a potential deletion of the first exon of this gene, including the start codon, supporting the notion of an *ASIP* LOF underlying the Tasmanian devil's melanistic coat color.

To confirm that the putative *ASIP* exon 1 deletion was not unique to the individual animal used to produce the mSarHar1.11 assembly, we next extracted the orthologous genomic region from another Tasmanian devil genome assembly available on NCBI, (SarHar_Dovetail_2.0), which was generated from a different animal and used different sequencing and assembly approaches. Additionally, we extracted this region from a third dasyurid species, the yellow-footed antechinus (AdamAnt_v2) for comparison. Interestingly, alignment of these regions revealed that the first, upstream region which had shown zero Tasmanian devil read mapping coverage likely represents an eastern quoll-specific insertion, as this sequence was not found in either Tasmanian devil or yellow-footed antechinus (Supplementary Data 2). However, consistent with our previous observations, our alignments also revealed identical deletion breakpoints for the putative deletion region encompassing *ASIP* exon 1 in both Tasmanian devil individuals (Supplementary Data 2).

Taken together, these results provide strong evidence for the fixation of a loss-of-function mutation in the Tasmanian devil ortholog of *ASIP*. Given the known effects of convergent *ASIP* LOF in other mammals, such a mutation is expected to cause a melanistic (non-agouti) pigment phenotype, thus underpinning the stark difference in background coat color between the Tasmanian devil and quolls.

Here, we present new, high-quality genomic resources for the endangered eastern quoll. Our chromosome-scale genome assembly closely matches the known chromosome complement of the species which is conserved among dasyurid marsupials. The completeness and contiguity of the assembly exceeds that of most existing marsupial reference genomes as indicated by high BUSCO recovery and low gap percentage. Our comparative analysis of core pigmentation loci reveals the probable basis of pelage variation between the eastern quoll and its close relative the

Tasmanian devil, and genome annotations provided here represent a tool for future comparative studies. Moreover, we identify preliminary evidence of demographic declines, reinforcing the value of future population genomic studies aimed at defining diversity, population structure, and genetic load in this species.

## Materials and methods

### Eastern quoll sightings map
An approximation of the eastern quoll's range within Tasmania was produced by accessing curated sighting data from studies recorded in the Tasmanian Natural Values Atlas (https://www.naturalvaluesatlas.tas.gov.au/), filtering to those within the 50-year period from 1/1/1973 to 1/1/2023. Locations were visualized on a map of Tasmania using a custom R script which is provided in a FigShare repository (https://figshare.com/s/29cdab4b5d5cde2c4d55).

### Tissue sampling
Samples of kidney, heart, liver, and spleen tissue were opportunistically collected as secondary use during the necropsy of an adult, brown morph female quoll named Manda, originating from the Aussie Ark captive breeding sanctuary. Samples were collected under the auspices of the Taronga Conservation Society Australia's Animal Ethics Committee (approval no. 3b1218), pursuant to NSW Office of Environment and Heritage-issued 543 scientific license no. SL100104. Excess tissue samples from this individual were deposited in the Australian Museum along with the pelt and skeleton under accession number M.52159. Subsampled tissues were snap-frozen and were stored at −80 °C until used.

Additionally, slices of the toe pad were taken and used to isolate primary dermal fibroblasts in culture. Briefly, small pieces of footpad tissue were scored into the base of a 30-mm plastic culture dish. 0.5 mL of DMEM (Gibco 10569044) containing 10% fetal bovine serum and antibiotic/antimycotic) was carefully added to the dish without disrupting the tissue pieces and incubated at 33 °C in 5% $CO_2$. The next day, medium was carefully added up to a total of 2 mL, then replaced every two days. Once the fibroblast outgrowths were 90% confluent, cells were expanded by trypsinization to a T25 flask, then five aliquots cryopreserved in fibroblast medium containing 10% DMSO. Cells were expanded in culture for eight passages before pellets were collected and frozen at −80 °C for RNA-seq.

### Genome sequencing and assembly
Subsamples of liver tissue originating from a single animal were used by Dovetail Genomics (Scotts Valley, CA USA) for genome sequencing and assembly. High molecular weight DNA was extracted using the Qiagen Blood & Cell Culture DNA miniprep kit (catalogue no: 13323) and quantified using a Qubit 2.0 Fluorometer (Life Technologies, Carlsbad, CA, USA). PacBio (Menlo Park, CA, USA) SMRTbell libraries were constructed using the SMRTbell Express Template Prep Kit 2.0. Libraries were bound to polymerase using the Sequel II Binding Kit 2.0 and were sequenced on Sequel II 8 M SMRT cells. This yielded ~97.68 gigabases of sequence comprised of 5,959,830 HiFi reads with an average length of ~16,390 nt. HiFi reads was then assembled into contigs using Hifiasm v0.15.4-r343 with default parameters[49]. Unpurged duplicates were then removed using purge_dups (v1.2.5) using automated thresholds[50].

A 3D chromatin contact library was produced from additional liver subsamples using Dovetail's Omni-C kit and was sequenced on an Illumina HiSeqX, generating approximately 126 gigabases of sequence (~420 million read pairs in 2 × 150bp format). Chromosome-scale scaffolds were then produced by first aligning Omni-C libraries to the de novo contigs with BWA mem v0.7.17 and the HiRise Pipeline was used to make contig joins and break putative misjoins[51,52]. Contact data were visualized by processing alignments with Pairtools (v1.0.2) and creating a .hic file with juicer_tools pre (v1.22.01) which was loaded into Juicebox (v1.11) and exported to generate a contact map[53–55].

### Genome assessment
Genome assembly metrics related to contiguity, base composition and contig/scaffold lengths were generated using the stats.sh script contained in the bbmap v39.01package[56]. Assembly completeness was further assessed via the recovery of benchmarking orthologs using BUSCO v5.4.6 in genome mode with the mammalia_odb10 database[57]. BUSCO gene recovery in the eastern quoll assembly was compared to that of all other marsupial whole-genome assemblies hosted on the NCBI Genomes database and labelled as the representative genome for their species as of April 18th 2023[21,22,28,29,46–48,58–61].

### Inference of chromosome homology
Homology between eastern quoll chromosomes and those of the related Tasmanian devil (*Sarcophilus harrisii*) and yellow-footed antechinus (*Antechinus* flavipes) were inferred using gene annotation overlap. Briefly, homologs of Tasmanian devil genes and antechinus genes respectively were identified via liftover to the eastern quoll genome using Liftoff v1.6.3 (parameters -d 4 -a 0.9 -s 0.9)[62]. Annotations were compared between chromosome-scale scaffolds in the eastern quoll and each reference dasyurid's genomes. Chromosomes sharing >= 95% of their gene content were deemed to be homologous. Annotation-based homology assignment was supported by the nearly identical relative sizes of presumptive homologous chromosomes between species, consistent with the exceptionally conserved karyotype previously reported among all examined dasyurids[23–26].

### RNA-sequencing and gene annotation
Frozen samples of eastern quoll heart, kidney, liver, spleen and a pellet of cultured dermal fibroblasts were provided to Psomagen Inc (Rockville, MD USA). RNA-seq libraries were prepared using the TruSeq Stranded mRNA LT sample prep kit (15031047 Rev. E) and sequenced on an Illumina NovaSeq 6000 in 2 × 150bp format. Residual adapters were removed, and reads were trimmed and filtered for quality using Trimmomatic v0.39 (parameters: SLIDINGWINDOW:5:15, MINLEN:50, AVGQUAL:20, ILLUMINACLIP:2:30:10)[63]. After processing, libraries ranged from approximately 29 to 40 million retained read pairs.

To annotate genes in the eastern quoll genome, filtered RNA-seq reads from all five tissues together with RefSeq homologous proteins from seven other marsupial species were provided to the Funannotate v1.8.14 pipeline which integrated these with Augustus ab initio predictions to infer gene models[64]. Within the Funannotate pipeline, the following modules were used: train (parameters: --no_trimmomatic, --stranded RF), predict (--augustus_species human, --busco_seed_species human, --optimize_augustus, --busco_db mammalia, --ploidy 1, --organism other, --min_intronlen 10, --max_intronlen 100000, --repeats2evm), and update (default parameters). This approach produced models for 29,622 genes, including 31,319 protein-coding transcripts. Of these, we were able to assign gene symbols to 14,293, based on high-confidence 1-to-1 orthology inferred using eggNOG-mapper's Mammalia database, a figure comparable to that of the Tasmanian devil and yellow-footed antechinus annotations produced by RefSeq (15,613 and 15,573, respectively)[65]. The density of annotated genes was visualized using the RIdeogram, providing a histogram of gene counts across 10 megabase (mb) windows on each chromosome[66].

### Repeat masking and annotation
Repeats in the eastern quoll assembly were annotated with RepeatMasker (v4.1.3) using the NCBI BLAST derived search engine rmblast, sensitive settings (-s), and a combined Dfam (v3.6) and Repbase (v20181026) repeat database for marsupials (-species metatheria)[30,67]. The repeat annotations produced were used to hardmask the assembly using Bedtools (v2.29.0)[68]. Subsequently, RepeatMasker was performed on the hardmasked assembly using a custom repeat library to identify KERV long terminal repeat elements (LTRs) and other marsupial-derived satellites absent in the above repeat databases. The resulting repeat annotations were combined and summarized using the RepeatMasker utility script buildSummary.pl.

## Demographic history

Haplotype-phased variants for the reference eastern quoll were identified by parsing aligned Omni-C reads with samtools v1.11 and pairtools and passing them to the Google Deepvariant (v1) pipeline which was run with default settings to call and filter variants[53,69,70]. Variant phasing was performed using HapCUT2 (v1.2)[71]. The three largest chromosomes in the eastern quoll genome exceed the maximum scaffold size limitations in the .bai alignment index required for several steps in data processing. Therefore, a second copy of the assembly was made, in which each of these scaffolds was split into two equal halves using samtools faidx. HiFi reads were then mapped against this copy using minimap2 v 2.24-r1122 (parameters -a -x map-hifi) and filtered with samtools view (parameters -h -q 20 -F 2304)[70,72]. The alignment file was then reduced to only chromosome-scale scaffolds corresponding to the six autosomes to prevent sex chromosome bias in recombination rate estimates. The average mapping depth was calculated with samtools depth and provided to the bamcaller.py included in the MSMC2 package to generate a mapping coverage mask file. Additionally, the RepeatMasker bed file was used to construct a mappability mask which excluded repetitive regions. These masks were provided together with heterozygous SNPs with QUAL > = 10 to the msmc-tools script generate_multihetsep.py (https://github.com/stschiff/msmc-tools). Finally, MSMC2 v2.1.4 was used to infer the eastern quoll's demographic history with 50 EM-iterations (−i 50) and time pattern (-p) $1*2 + 20*1 + 1*5$, reducing the number of free parameters relative to default settings and grouping the last five time segments to reduce overfitting at extremely ancient time periods. Data were then plotted against climatic changes[32] using the previously-reported generation time of two years and a per-site, per-generation mutation rate of 5.95e-9[36,37].

## Comparative genomics

The *ASIP* sequence was annotated in all available dasyuromorph reference genomes on NCBI via lift-over from the annotation of the yellow-footed antechinus (see: 'Inference of chromosome homology'). Sequences were then extracted with gffread, translated into the conserved reading frame using MACSE v2, and then aligned using the MAFFT web server using default parameters[73–75].

To generate coverage histograms, Tasmanian devil long reads used to generate mSarHar1.11 were accessed from NCBI SRA (Accession: ERR3930603) and aligned against the eastern quoll genome with minimap2 v2.24 (parameter: -x map-ont). Pileups of reads mapped to the *ASIP* locus were then visualized using IGV[76].

To perform alignments of genomic sequence surrounding the *ASIP* exon 1 region, *ASIP* exon 2 from the Tasmanian devil was aligned against the eastern quoll (DasViv_v1.0), yellow-footed antechinus (AdamAnt_v2), and Tasmanian devil (mSarHar1.11 and SarHar_Dovetail_2.0) assemblies using blastn with default settings[77]. Samtools faidx was then used to extract the region containing the putative deletion region (i.e., between conserved flank sequences found in all species). These regions were then aligned using the MAFFT web browser with default parameters[70,75].

## Reporting summary

Further information on research design is available in the Nature Portfolio Reporting Summary linked to this article.

## Data availability

The eastern quoll reference genome and all sequence data used in its generation are available on NCBI under BioProject PRJNA758704. Transcriptome data used to produce gene annotations are available under BioProject PRJNA963007. Gene annotation GFF files and all original code used in this study can be accessed in a permanent FigShare repository: https://figshare.com/s/29cdab4b5d5cde2c4d55.

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

## Acknowledgements

We thank Revive & Restore for their organizational support for this project with special thanks to Bridget Baumgartner, Cantata Bio LLC for computational support and sequencing services with special thanks to Mark Daly and Jordan Zhang, Karrie Rose for collecting tissues, Sandy Ingleby and Emma Peel for depositing samples at Australian Museum, the Computational Biology Core in the Institute for Systems Genomics at the University of Connecticut for the use of the Xanadu HPC, Brett Vercoe for the use of his eastern quoll and Tasmanian devil photographs and Elise Ireland for proofreading the manuscript. This project was supported by a Wild Genomes grant to C.Y.F., A.J.M., N.M.R., R.K.H., C.P.B. and M.E.J. from Revive & Restore (Contract no. 2020-017). C.Y.F. was supported by NIH NRSA fellowship F32GM139240-01. G.A.H. and R.J.O. are supported by a grant from the US National Institutes of Health R01GM123312-02.

## Author contributions

C.Y.F., A.J.M., N.M.R., R.K.H., C.P.B. and M.E.J. conceived the study and acquired funding. C.Y.F. performed assembly quality, gene annotation, demographic history, and comparative genomic analyses. G.A.H. performed genomic repeat characterization. S.R.F. isolated dermal fibroblasts. R.B., T.F., H.S. K.R. and N.M.R. M.B.R., R.O. and A.J.P. acquired samples and undertook sample logistics. C.Y.F., G.A.H. and S.R.F. wrote the manuscript with input from all authors.

## Competing interests

The authors declare no competing interests.
