## [Peer Review File · Communications Biology]

Reviewers' comments:

Reviewer #1 (Remarks to the Author):

General points:

1) What are the major claims of the paper? Are they novel and will they be of interest to others in the community and the wider field?

The authors present a chromosome-level reference assembly and multi-tissue transcriptome - supported annotation of an endangered marsupial, the eastern quoll. Using comparative genomics, they identify a deletion in the ASIP gene, potentially responsible for the specific pelage phenotype of the eastern quoll.

The development of these new genomic resources are not only important for the conservation and management of the endangered species, but also for further comparative genomic studies in marsupials.

2) The manuscript is technically and conceptually sound, well written and informative.

A few questions/open points remain though.

a) As this is a manuscript addressing genomic/ genetic topics, in the introduction previous genetic studies are missing. It would be a reasonable assumption that eastern quolls have been investigated before using mtDNA or nuclear markers and there might be information about genetic diversity, population structure, demography, etc. This needs to be included in the introduction. Maybe, in earlier studies specific genetic questions were raised that make the development of the reference genome necessary. It would be interesting to link this study to earlier (conservation) genetic investigations on the species.

b) What were the aims of the study beyond establishing a reference genome? If it was "only" to provide a reference genome (resource), then a more specific journal might be a better place for the manuscript. It would be interesting to read into the introduction the other targets of the manuscript beyond providing a new reference genome, e.g. comparative genomics, demography, etc.

c) The homology analysis was performed with Tasmanian devil and yellow-footed antechinus but not with the parametric spotted-tailed quoll or other marsupial genomes, why?

3) Are the conclusions original? If not, please provide relevant references. Is the work convincing, and if not, what further evidence would be required to strengthen the conclusions?

The conclusions of the manuscript are original, the work is convincing and the provided evidence mostly supports the conclusions, except:

Line 307-308: "Only" related Tasmanian devil and yellow-footed antechinus were used for the chromosome homology analysis, but no other available marsupial genomes. Please either add more genomes to the analysis, or reconsider the conclusions.

The discussion in general is very short and reads more like a conclusion paragraph.

The discussion needs to be improved, including the application of the genomic resource for conservation management (as proposed in the introduction), phylogenetic relationship between the new reference genome of the eastern quoll and other related whole-genomes, transcriptomic analysis, etc.

4) Do you feel that the results presented are of immediate relevance for people in your own discipline or for a broader audience? If you recommend publication, please outline briefly what you consider to be the outstanding features.

The results presented in the manuscript are of high interest for scientists in the field as well as for a broader audience.

Outstanding features of the manuscript are:

The chromosome-level assembly of the endangered eastern quoll is of high quality.

The annotation is based on multi-tissue transcriptome analysis.
Demographic analysis of the endangered eastern quoll shows historic population decline.

5) If you feel that specific additional experiments would strengthen the case for publication in *Communications Biology*, please provide suggestions.

The comparative genomic analysis is limited to the ASIP locus. It is not clear why the authors a priori chose this gene and did not extend their analysis to a) more genomic regions, b) more marsupial genomes.

It would be interesting to see a phylogenomic analysis with other available marsupial genomes as provided in lines 111-113. For an example of such analysis, e.g., see doi: 10.1111/1755-0998.13654. What happened with all the transcriptomic data? It would be very interesting to see the results of the transcriptomic analysis beyond the usage for annotation.

6) Appropriateness and validity of any statistical analysis, as well the ability of a researcher to reproduce the work, given the level of detail provided.

The bioinformatic analysis is appropriate and valid. The original code used in the study has been provided in the data availability statement. All data have been made available.

Specific points:

Fig.1: Please can you provide the source of the 50 years of eastern quoll sightings (blue data points) in the legend, or was it part of this study?

l68: Please add here a note for the availability of the R script at fig share (in addition to the data availability statement).

l192: repetition of "were then", please delete

195-214: eastern quoll reference genome: Please can you provide the most important features of the reference genome in a table, including size, total sequence length, total sequence number, average sequence length, GC content, N50, N90, etc.

l227-257: Historical demography: a) A demographic analysis is always historical (the development of a population over time). Maybe choose a more informative subtitle. b) This paragraph is a mix between results and discussion. While I have no problem with presenting results and discussion together, this should follow the journal's style and also be applied consequently throughout the manuscript. I.e., separated result and discussion sections or a combined results & discussion section. But it should not change between results paragraphs (one paragraph only showing results, while the other is a mix between results and discussion).

Reviewer #2 (Remarks to the Author):

The "Genome of the endangered eastern quoll (*Dasyurus viverrinus*) reveals signatures of historical decline and pelage color evolution" describes the sequencing and assembly of a high-quality genome for the eastern quoll, an endangered marsupial. The authors use the latest sequencing and assembly methods and the assembly looks good by all available metrics. The authors compare this assembly to other dasyurids genomes and suggest the ASIP locus is responsible for the dark coloration of Tasmanian Devils. Overall this was a well-written paper and the results support the conclusions. I have some relatively minor comments (listed below) that I hope the authors find useful.

Line 88 – "Samples of liver tissue..." – Were these all from the same individual mentioned in the tissue sampling section (M.52159)? Perhaps add a sentence to the previous section clarifying, "all DNA and RNA sequencing was performed on samples from this individual."

Lines 110-113 – was BUSCO run on all the other marsupial genomes or were BUSCO values taken from the publications from those respective genomes. The authors should be clear what they did here. Worth noting that different versions of BUSCO will give different summary statistics so it is preferred that BUSCO be run on everything using the same software version when making comparisons among assemblies.

115-116 – Include the scientific name of the antechinus. The Tasmanian devil scientific name was included earlier in the manuscript and does not need to be repeated here.

Line 165 → Why was the X chromosome excluded?

Lines 168-169 – " Additionally, the RepeatMasker bed file was used to identify construct a mappability mask which excluded repetitive regions" Is "identify" needed here?

Line 192 – remove "were then and"

Figure 3 – What are DNA repeats (in red on the bar plot)? Is this the DNA transposon category from RepeatMasker? Perhaps there is a better way to label this?

Reviewer 1:

The authors present a chromosome-level reference assembly and multi-tissue transcriptome - supported annotation of an endangered marsupial, the eastern quoll. Using comparative genomics, they identify a deletion in the ASIP gene, potentially responsible for the specific pelage phenotype of the eastern quoll. The development of these new genomic resources are not only important for the conservation and management of the endangered species, but also for further comparative genomic studies in marsupials. The manuscript is technically and conceptually sound, well written and informative. A few questions/open points remain though.

We thank the reviewer for their comments on our manuscript and for their appreciation of the scientific and practical value of the work and resources presented. Please find below our responses and descriptions of amendments in the updated manuscript.

As this is a manuscript addressing genomic/ genetic topics, in the introduction previous genetic studies are missing. It would be a reasonable assumption that eastern quolls have been investigated before using mtDNA or nuclear markers and there might be information about genetic diversity, population structure, demography, etc. This needs to be included in the introduction. Maybe, in earlier studies specific genetic questions were raised that make the development of the reference genome necessary. It would be interesting to link this study to earlier (conservation) genetic investigations on the species.

We appreciate the reviewer's comments about the importance of contextualizing our study with previous conservation genetics studies performed with the eastern quoll. We have added text to the introduction highlighting previous genetic work using low resolution microsatellite markers, the limitations of such prior studies and the value of our reference genome on Lines 60-63 in the revised manuscript.

What were the aims of the study beyond establishing a reference genome? If it was "only" to provide a reference genome (resource), then a more specific journal might be a better place for the manuscript. It would be interesting to read in the introduction the other targets of the manuscript beyond providing a new reference genome, e.g. comparative genomics, demography, etc.

We agree that the focus of our subsequent analyses should come through more clearly in the introduction. We have amended the text accordingly on Lines 60-68 in the revised manuscript.

The homology analysis was performed with Tasmanian devil and yellow-footed antechinus but not with the parametric spotted-tailed quoll or other marsupial genomes, why?

We direct the reviewer to the below comment, where we have elaborated on this point.

**The conclusions of the manuscript are original, the work is convincing and the provided evidence mostly supports the conclusions, except:
Line 307-308: "Only" related Tasmanian devil and yellow-footed antechinus were used for the chromosome homology analysis, but no other available marsupial genomes. Please either add more genomes to the analysis, or reconsider the conclusions.**

We thank the reviewer for their positive comments on the originality and credibility of our work. However, additional homology analyses were not performed on other dasyurids, including the (parapatric) spotted-tailed quoll because a chromosome-level reference genome for this species has not been published. The Tasmanian devil and yellow-footed antechinus, both used in this study, are the only other full chromosome-level assemblies available for dasyurids. While other assemblies exist, including dasyurids and outgroups (numbat and thylacine), these are not chromosome-level (or in the case of the thylacine have reference-guided scaffolds). Consequently, these are unsuitable for use in identifying chromosomes in our assembly via homology. We would also like to note that, while a "reference genome" is available on NCBI for the spotted-tailed quoll, this entry is not actually a genome assembly. It is instead part of a large set of marsupial partial exome capture experiments (each representing <1% of the genome), which were erroneously updated as representative genomes and which have not yet been corrected. One of our key motivations for generating an eastern quoll assembly is the paucity of dasyurid genomes available.

Aside from the unavailability of dasyurid genomes suitable for identifying 1-to-1 chromosome homology, we wish to clarify the intent of the analysis and the nature of its conclusion, which the reviewer points out on lines 307-308 of the original manuscript:

"Our chromosome-scale genome assembly closely matches the known chromosome complement of the species which is conserved among dasyurid marsupials"

That the eastern quoll shares the same conserved karyotype as all other known dasyurids is not a finding of our study, but it was already established through previous cytogenetic research. Our goal was to check that the chromosome-scale scaffolds in our assembly match the already-known eastern quoll/dasyurid karyotype and to assign their homology. Nevertheless, we thank the reviewer for pointing this out and in an effort to be clearer about our conclusions, we have added further citations to Lines 132 and 214 of the revised manuscript to highlight relevant cytogenetics work.

The discussion in general is very short and reads more like a conclusion paragraph. The discussion needs to be improved, including the application of the genomic resource for conservation management (as proposed in the introduction), phylogenetic relationship between the new reference genome of the eastern quoll and other related whole-genomes, transcriptomic analysis, etc.

We agree with the reviewer that the original contents of this section would have been better described as 'Conclusions' rather than 'Discussion'. We believe that our current 'Discussion' section, an optional section for *Communications Biology* would be better suited as closing remarks of a combined 'Results and Discussion' section. Accordingly, we have modified these section headers. Additionally, we have incorporated brief commentary on the application of the genomic resources presented in the corresponding subsection on Lines 224-226 of the revised manuscript.

The results presented in the manuscript are of high interest for scientists in the field as well as for a broader audience. Outstanding features of the manuscript are: The chromosome-level assembly of the endangered eastern quoll is of high quality. The annotation is based on multi-tissue transcriptome analysis. Demographic analysis of the endangered easter quoll shows historic population decline.

We are very grateful for the reviewer's interest in our manuscript and their kind words about its value.

If you feel that specific additional experiments would strengthen the case for publication in Communications Biology, please provide suggestions. The comparative genomic analysis is limited to the ASIP locus. It is not clear why the authors a priori chose this gene and did not extend their analysis to a) more genomic regions, b) more marsupial genomes. It would be interesting to see a phylogenomic analysis with other available marsupial genomes as provided in lines 111-113. For an example of such analysis, e.g., see doi: 10.1111/1755-0998.13654. What happened with all the transcriptomic data? It would be very interesting to see the results of the transcriptomic analysis beyond the usage for annotation.

We appreciate the reviewer's suggestions and wish to clarify that our *a priori* reasons for examining ASIP are provided on Lines 266-274 in the original manuscript (Lines 278-286 in the revised manuscript).

We agree that transcriptomic analyses are a powerful tool to study the unique aspects of a species' biology. However, the samples and sequencing design that we chose herein (single samples from diverse tissues that were sequenced to relatively high depth) were required for optimal gene annotation. We wholeheartedly agree that moving forward, we should strongly consider cases where an appropriately designed and question-driven transcriptomic experiment could advance aspects of our work (e.g. differential expression with proper replicates and tissues/conditions of interest, clustering or network analyses focused on an aspect of quoll development, or total RNA libraries (instead of mRNA) suitable for examining more variable classes of transcripts like non-coding RNAs).

The validity of the method underlying CAFE (the program used in 10.1111/1755-0998.13654 to analyse gene family expansion/contraction) has been brought into question by the results of Louca & Pennell (<https://doi.org/10.1038/s41586-020-2176-1>). Their work showed that for any analysis employing generalized birth-death rate models, an infinite number of highly disparate but equally likely lineage histories exist. Regarding phylogenetic analyses, the relationships between the eastern quoll and other marsupials with available genome assemblies are well established and divergence time estimates have been explored elsewhere. In the future as more marsupial genomes become available (particularly other quoll species, which have disputed relationships), we would be very interested in conducting novel phylogenetic analyses.

We note that our manuscript is comparable in depth/breadth to other recent papers in Communications Biology (e.g. <https://doi.org/10.1038/s42003-022-04074-5>) and are grateful for the very clear agreement articulated by our reviewers on the value and broader impact of both the resources we have generated and our specific findings. We thank the reviewer for suggesting analyses which will be of considerable interest in our future work.

Specific points:

Fig.1: Please can you provide the source of the 50 years of eastern quoll sightings (blue data points) in the legend, or was it part of this study?

These data are from the Tasmanian Natural Values Atlas, a government database. This information can be found in the Figure 2 legend and in the first paragraph of the methods section (Lines 72-75 in the revised manuscript) with a link to the database, along with the dates between which the observations occurred.

I68: Please add here a note for the availability of the R script at fig share (in addition to the data availability statement).

Thank you for this suggestion. We have added a note on the availability of the R script in the FigShare repository as indicated on Line 75 in the revised manuscript.

I192: repetition of "were then", please delete

We thank the reviewer for catching this typo. We have amended it.

195-214: eastern quoll reference genome: Please can you provide the most important features of the reference genome in a table, including size, total sequence length, total sequence number, average sequence length, GC content, N50, N90, etc..

This information was included as a supplemental table in the original manuscript (Supplementary Table 2); however, we agree with the reviewer that such a table would be more accessible in the main text. In the revised manuscript we have moved Supplementary Table 2 into the main text as Table 1.

I227-257: Historical demography: a) A demographic analysis is always historical (the development of a population over time). Maybe choose a more informative subtitle. b) This paragraph is a mix between results and discussion. While I have no problem with presenting results and discussion together, this should follow the journal's style and also be applied consequently throughout the manuscript. I.e., separated result and discussion sections or a combined results & discussion section. But it should not change between results paragraphs (one paragraph only showing results, while the other is a mix between results and discussion).

We agree, all demographic analyses are historical (though not all histories are demographic in nature). The authors of MSMC were more precise in making this distinction. In the amended text we have changed all occurrences of "historical demography" to "demographic history" to match the phrasing used in the publications associated with MSMC.

We wish to thank the reviewer again for their time, consideration, and comments on our manuscript.

Reviewer #2 (Remarks to the Author):

The "Genome of the endangered eastern quoll (*Dasyurus viverrinus*) reveals signatures of historical decline and pelage color evolution" describes the sequencing and assembly of a high-quality genome for the eastern quoll, an endangered marsupial. The authors use the latest sequencing and assembly methods and the assembly looks good by all available metrics. The authors compare this assembly to other dasyurids genomes and suggest the ASIP locus is responsible for the dark coloration of Tasmanian Devils. Overall this was a well-written paper and the results support the conclusions. I have some relatively minor comments (listed below) that I hope the authors find useful.

We thank the reviewer for their constructive comments and their positive view of our work.

Line 88 – "Samples of liver tissue..." – Were these all from the same individual mentioned in the tissue sampling section (M.52159)? Perhaps add a sentence to the previous section clarifying, "all DNA and RNA sequencing was performed on samples from this individual."

Thank you for noting that this is indeed unclear. Yes, all tissues in this paper are from the same individual. We have amended the text in this section accordingly (Line 95 in the revised manuscript) and also replaced "samples" with "subsamples".

Lines 110-113 – was BUSCO run on all the other marsupial genomes or were BUSCO values taken from the publications from those respective genomes. The authors should be clear what they did here. Worth noting that different versions of BUSCO will give different summary statistics so it is preferred that BUSCO be run on everything using the same software version when making comparisons among assemblies.

These are indeed all newly-run BUSCO analyses with the same version. Note that we have also included a new supplemental (Supplementary Table 3 in the revised manuscript), which contains the raw BUSCO results for all species.

115-116 – Include the scientific name of the antechinus. The Tasmanian devil scientific name was included earlier in the manuscript and does not need to be repeated here.

We have added the species names as indicated.

Line 165 – Why was the X chromosome excluded?

We thank the reviewer for asking about this. MSMC2 attempts to estimate the genome-wide recombination rate as part of the algorithm. However, the mammalian X and Y chromosomes have very different recombination rates compared to the autosomes due to hemizyosity in males. Thus, the rate calculated for the genome can either be skewed by sex chromosome data or vice versa, which could (at least in principle) affect estimated coalescence times.

We do note that in the particular case of the dasyurids, which have a small X chromosome and enormous autosomes, the whole X only corresponds to about 3% of the genome. This is far less than what is excluded in a typical analysis due to mappability and coverage masking. The tiny size of the X renders its removal unnecessary in practice, but we still applied it on first principles. For clarity we have amended our methods to include this rationale on Lines 173-175.

Lines 168-169 – " Additionally, the RepeatMasker bed file was used to identify construct a mappability mask which excluded repetitive regions" Is "identify" needed here?

Line 192 – remove "were then and"

We thank the reviewer for catching these two typos. We have removed "identify" and the extra "were then and".

Figure 3 – What are DNA repeats (in red on the bar plot)? Is this the DNA transposon category from RepeatMasker? Perhaps there is a better way to label this?

Yes, this category reflects the DNA transposon category in RepeatMasker. We have edited the figure and Supplementary Table 5 in the revised manuscript to reflect this.

We appreciate the reviewer's attention to detail and for their comments on our manuscript.

REVIEWERS' COMMENTS:

Reviewer #1 (Remarks to the Author):

I thank the authors for implementing all my suggestions and I do not have no further comments.